# Probability of Risk Factors Affecting Small-Scale LNG Supply Chain Activities in the Indonesian Archipelago

Rossy Armyn Machfudiyanto [1], Windra Priatna Humang [2,*], Nurul Wahjuningsih [3], Insannul Kamil [4] and Yanuar Yudha Adi Putra [1]

1 Department of Civil Engineering, Faculty of Engineering, Universitas Indonesia, Kampus Baru UI Depok, Depok 16424, Indonesia
2 Research Center for Transportation Technology, National Research and Innovation Agency, Tangerang Selatan 15314, Indonesia
3 Faculty of Civil and Environmental Engineering, Institut Teknologi Bandung, Bandung 40132, Indonesia
4 Department of Industrial Engineering, Faculty of Engineering, University of Andalas, Kampus Limau, Padang 25163, Indonesia
* Correspondence: windra.priatna.humang@brin.go.id

**Abstract:** In Indonesia, the CoS for power supply increased from Rp. 1025 to Rp. 1334/KWh from 2016 to 2021, respectively; this indicates an inefficient process in electricity provision. One contributing factor to this inefficiency is the existence of many high speed diesel (HSD)-fueled power plants. These are distributed across the Indonesian archipelago with a supply chain that only uses sea transportation. The absence of an economical small-scale LNG (SS-LNG) supply chain also demonstrates the inadequate infrastructure for distributing LNG to refineries. This study aims to analyze the probability of risks that occur in SS-LNG supply chains in the Indonesian archipelago. The analytical methods used are descriptive statistical analysis and Delphi analysis through in-depth interviews and Focus Group Discussions (FGD) with experts. Results showed that the SS-LNG supply chain process in Indonesia includes LNG loading, unloading, shipping, picking, storage, regasification, and distribution. There are 30 risk indicators that may occur, with the highest risks including ship accidents, equipment damage, lack of transport ships, bad weather, earthquakes, tsunami, poor safety cultures, and low levels of safety leadership. These risk indicators can be used in implementing SS-LNG.

**Keywords:** archipelago; risk; Small-Scale LNG; supply chain





## 1. Introduction

The Indonesian government's request through the Ministry of Energy and Mineral Resources emphasizes the improvement of electricity supply efficiency by PT PLN (Persero). This request shows that efficiency is a component parameters used in calculating the Cost of Supply (CoS) and electricity subsidy needs. In Indonesia, the significant components of CoS are the electricity (power generation), network, and other operating costs, which are 72%, 11%, and 17%, respectively [1]. From 2016 to 2021, the CoS reportedly increased from 1025 to 1334 IDR/KWh, indicating that the country's increasingly inefficient power generation caused the cost of supply to increase [2].

Figure 1 shows the comparison of Indonesian CoS to those in any location of the archipelago, demonstrating that almost all the CoS in any area with power plants is above the country's average, especially in the central and eastern regions. In the archipelago, especially in Central and Eastern Indonesia, the high CoS value displays the inefficiency of electricity generation in the area, which impacts the inadequacy of the national average power CoS. Based on PLN's RUPTL (2016–2025), 30 PLTG and PLTMG were reported; these used gas or oil fuel engines with fuel-dueling designs.

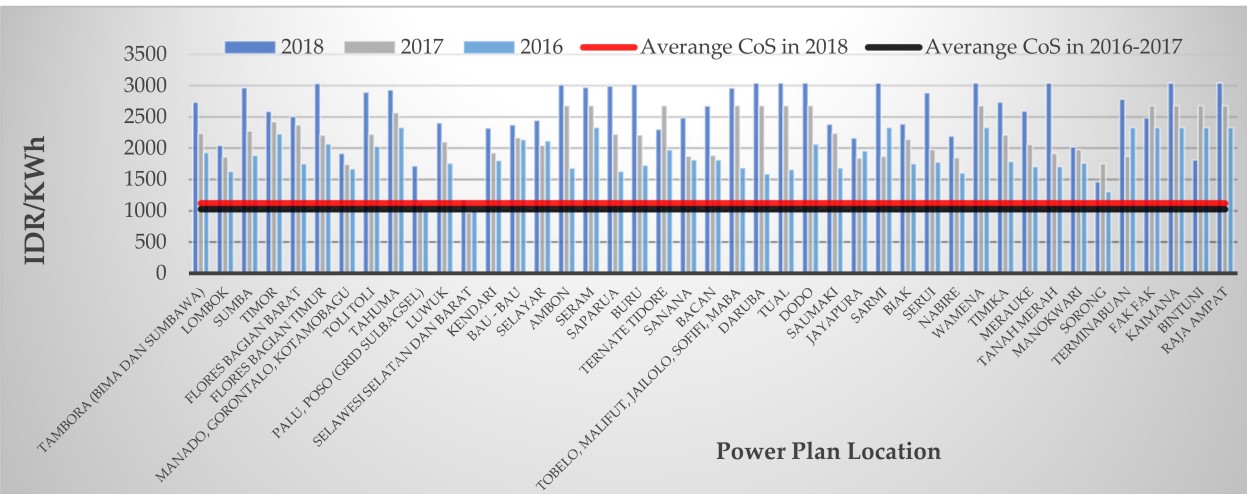

**Figure 1.** Comparison of Indonesian CoS to CoS Per location in Indonesia [3–6].

The price comparison of the archipelagic Gasoil CN 48 or High Speed Diesel (HSD) to LNG showed the highest difference (65%) in 2016 and lowest difference (16%) in 2013. This indicates that LNG prices were frequently below the prices of HSD, at an average of 38% from 2010 to 2020 [2]. Oil production yielded a downward trend from 346 to 283 million barrels (949 and 778 thousand BPD) from 2009 to 2018, respectively [7]. This decline was due to the age of the main petroleum production wells, with the yield of the new sources relatively limited. Since Indonesia mainly imports oil from the Middle East to meet refinery needs, the importation dependence reached about 35% [7]. Using HSD in 30 existing and 22 potentials (2022) PLTG and PLTMG, respectively, several effects are observed in the country, such as higher dependence on petroleum imports [7]; higher electricity CoS, especially in the archipelago [4–6]; increase in the value of electricity subsidies by the state, which amounted to 45.74 and 53.59 trillion Rupiah in 2017 and 2021, respectively [3,8]; increase in the electricity tariff for households from 93 USD/BOE in 2013 to 129 in 2019. The tariff decreased to 115 USD/BOE in 2020 because electricity demand declined during the COVID-19 pandemic [8].

HSD is often used in gas-fired power plants (PLTG/PLTMG) because LNG is not available. This emphasizes the existing operation with dual-fuel engines and leads to a problem/challenge from the LNG supply chain to power plants. Furthermore, there is a relatively small need for LNG/natural gas plants to replace HSD; the low and high of 0.68 BBTUD and 7.74 BBTUD, respectively, with an average need of 2.28 BBTUD [9]. According to the International Gas Union [10], two parameters commonly used as differentiators between small and large-scale LNG, namely LNGc capacity and reception tanks, which were developed for scales <30,000 m$^3$ and 500–30,000 m$^3$, respectively. Regarding the small-scale parameter, Setyorini [11] and Antara [12] also stated that LNG supply chain models for Kalimantan, Bali, Nusa Tenggara, and Papua need to use capacities below 30,000 m$^3$.

Figure 2 shows the LNG refinery, power plants, and wave height locations in Indonesia, which are distributed within the archipelago and are separated by the sea and ocean. Moreover, the next challenge in LNG supply chains is the sea depth conditions in coastal areas, where PLTG/PLTMG is relatively shallow [13]. The condition of the Indonesian sea and ocean included fierce waters, with wave heights reaching 4–5 m. According to the Shipping Court (2003–2013), this leads to a higher risk for sailing; indeed, 349 ship accidents were observed within the country, with 23% of them (82) caused by Force Majeure or Environmental Factors [14]. These hazardous conditions subsequently caused larger ships to be developed which were better able to handle coastal environmental situations and reduce the risk of marine accidents. Apart from these conditions, investment costs are have a significant influence on more expensive ships.

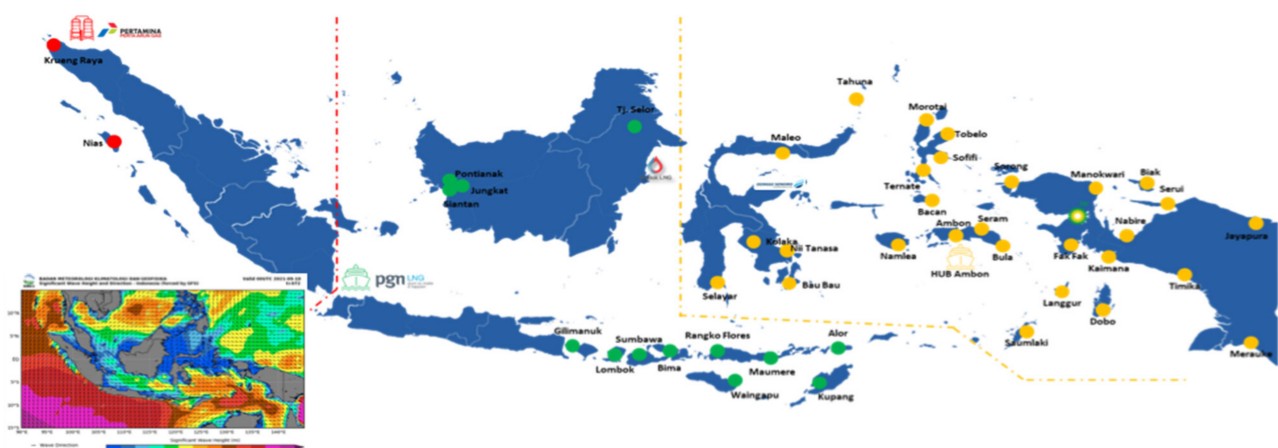

**Figure 2.** LNG Refinery Location, PLTG/PLTMG Location, and Wave Height [7,9,15].

A small-scale LNG (SS-LNG) supply chain has also been implemented for a 200 MW PLTDG plant in Pesanggaran Bali, Indonesia. This was the first small-scale terminal in Southeast Asia. The supply chain reportedly begins with the LNG sources obtained from the Bontang Plant, using a small transport ship loaded with 23,000 m$^3$ of LNG. The ship is then transported to the collection terminal in Benoa Bali; the terminal has a floating LNG storage capacity and regasification unit of 19,000 m$^3$ and 2 × 25 MMscfd, respectively. Gas distribution was also conducted, using a pipeline of 700 m [12,16]. Based on the Minister of Energy and Mineral Resources Regulation No. 45 of 2017, which concerns cost reduction by streamlining a plant's supply chain, the highest possible LNG plant prices should be 14.5% of ICP (Indonesian Crude Price) [17,18].

The SS-LNG supply schemes for Bali, Lombok, and Papua power plants have reportedly not met the economic feasibility and efficiency factors [13,19]; this prompted the Ministry of Energy and Mineral Resources to implement the Ministerial Decree No. 13/2020, which concerns the Implementation of LNG Infrastructure Supply and Development. In this case, an efficient risk-based small-scale LNG supply scheme is needed to obtain electricity CoS costs that are lower than those of HSD power plants or are a maximum of 14.5% of ICP.

To understand supply change flexibility, it is important to understand the relationship between environmental uncertainty and supply chain risk [20]. The risks and costs of having a very close relationship impact the high cost of the transportation and shipping insurance that is required because of the increasingly dangerous sailing line, especially in the SS-LNG supply chain [19]. The development of this model considers various risk factors, such as sea transportation distance, pirate attacks, import economic hazards, state political export dangers, and marine commutation disasters. According to Gurning [21], unexpected disruptions cause high costs and problems in supply chain systems, such as long waiting times, unavailable stock, inability to satisfy customers, and increased expenditure.

In one study, the relationship of the risk variables to the SS-LNG supply chain activities, such as the need for re-routes on logistics lines, caused increased distance between terminals. The ship's speed was also decreased, leading to lengthy voyage and docking periods. Supply levels were increased, which mitigated the risk of delayed delivery. The study's results also influenced higher investment and operating costs [22]. The risks in SS-LNG activities in Indonesia include human factors, environmental factors, and technical factors; these risks can result in increased cost and sailing time in the SS-LNG supply chain process.

Therefore, this study aims to identify the risks that may occur, and the probability of an impact occurring is based on the index of each risk. After the risk index is determined, mitigation can be prepared for all the risks that occurred. This study used the Delphi approach and risk index assessment as approaches to answering research objectives. The research results contribute to efforts to reduce costs associated with and the time necessary

to implement the SS-LNG supply chain. Future research will be conducted regarding the development of a risk-based SS-LNG supply chain model.

## 2. Theoretical Study

### 2.1. Small-Scale LNG Supply Chain

Figure 3 illustrates the four main elements of the LNG supply chain, namely (1) Sea/land gas wells (offshore operations), (2) LNG refinery (onshore production and storage) (3) Transportation (shipping operations), and (4) LNG collection terminal (delivery and consumption). These elements show that the supply chain is a natural gas network, in which gas is often transported from the gas field to the liquefaction refinery, where impurities or contaminants are removed. After the removal processes, the gas is cooled to −162 °C, transforming it to a liquid form that flows into the storage tank, which is then loaded on a carrier for transport to the collection terminal. At that point, the regasification process returns the product into a gas, which is then transported to end-users primarily through pipelines and ships. Based on recent analysis, LNG vessels are more in demand as a medium for transporting natural gas due to the high transportation cost and geographical limitations of pipeline distribution over long distances [11,12].

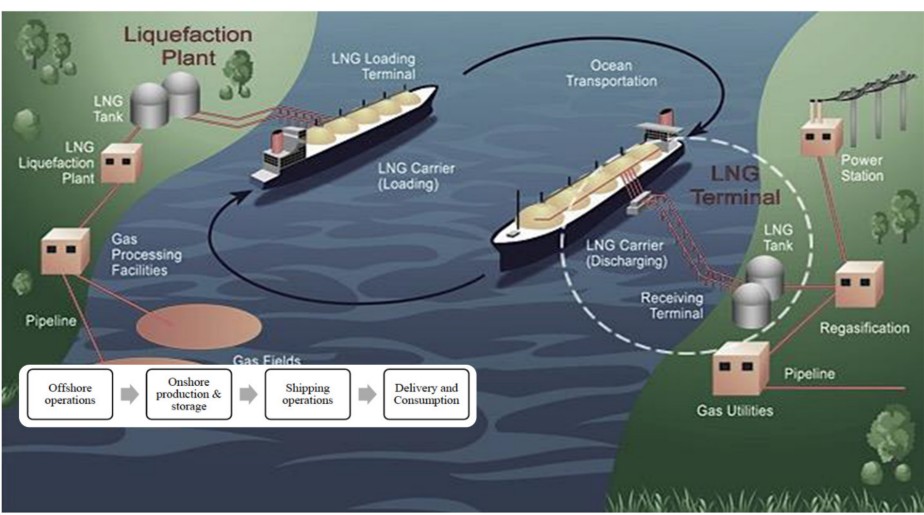

**Figure 3.** LNG Supply Chain [11,23].

Maritime transport challenges have unique practical and theoretical perspectives. From a practical perspective, this transport process is aligned with economies of scale, leading to it being promoted as the cheapest per unit option for the mobility of large industrial activities. Meanwhile, theoretical perception indicates that the optimal solution to maritime transport challenges within a reasonable time frame is potentially problematic [23]. Table 1 shows several main activities for each major element.

**Table 1.** LNG Supply Chain Activities.

| Variable/Code | Activities/Code |
|---|---|
| LNG Plant X.1.1 | LNG Loading, X.1.1.1 LNG stored in storage tanks is pumped to the product dock. |
| | X.1.1.2 At the dock, LNG is loaded onto the export vessels. |
| | X.1.1.3 Boil of Gas (BOG) formed on the tarmac is retransferred to the LNG plant. |
| | LNG Loading, X.1.1.4 Ship activities include the following (a) berth to the dock, and (b) unberth with stages of changing basins (turning basins), docking, as well as preparation for loading and unloading, and departure |

**Table 1.** *Cont.*

| Variable/Code | Activities/Code |
|---|---|
| Transportation X.1.2 | LNG Shipment, X.1.2.1 LNG-laden ship sails from LNG plant to the collection terminal X.1.2.2 Ships/trucks transport LNG from the first to the next terminals |
| | LNG Retrieval, X.1.2.3 The ballast condition vessel returns to the plant to obtain LNG |
| LNG Receiving Terminal X.1.3 | LNG Unloading, X.1.3.1 Removing LNG from the ship, using the pump and loading arm (unloading arm) at the dock, and returning the boil of gas (BOG) to the ship's tank keeps the pressure at 8–10 KPa. |
| | X.1.3.2 Ship activities include the following, (a) berth to the dock and (b) unberth with stages of changing the basins (turning basins), docking, loading and unloading preparations, as well as departure |
| | LNG Storage: X.1.3.3 Storing LNG in onshore tanks |
| | LNG Regasification: X.1.3.4 LNG is subjected to pressure using the pump in a tank, and then converted into gas. This is carried out by heating, through media such as seawater, hot water, and air. |
| | Gas Distribution, X.1.3.5 Addition of odorant (odorant) to gas, and gas delivery to customers/power plants through pipes |

Source: [23–25].

According to the International Gas Union [10], two parameters are commonly used as differentiators between small-scale and large-scale LNG: the LNGc capacity (<30,000 m$^3$) and collection tank (500–30,000 m$^3$). Alice Bittante and Saxen [26] also stated that SS-LNG used 1000–40,000 m$^3$ LNGc and <50,000 m$^3$ collection tanks. Furthermore, small-sized LNG vessels had a transportation cost per gas volume, which was more expensive than using larger ships with similar distance coverage. Irrespective of these conditions, the selection of small-sized LNG vessels still capitalized on the flexibility of depth requirements, cheaper rental costs, and conformity with the demand for small LNG supplies in the Indonesian archipelago.

*2.2. Risk and Cost*

Hazard identification begins by analyzing the functions and processes of a system. This is accompanied by identifying the potential hazards that pose systematic risks or losses. In this case, LNG supply chain operators often encounter various uncertainties in specific parts of the supply chain element as various internal and external factors can interfere with shipments between the source and the planned destinations. Moreover, a third-party strategy is commonly used for these elements, which often have high, dominant uncertainty [27]; these include related maritime transportation elements, namely, the transportation services downstream of supply chain operations. These uncertainties subsequently include the risks associated with LNG transport in shipping-related industries, such as: (1) shortage of fleets for one specific route, (2) natural sea hazard conditions at sea, such as severe wave height, and (3) ship supply/demand imbalances [21]. For Gurning [21], Mokhatab et al. [25], and Wan [28], some related categories were proposed, namely: (1) security and safety, (2) service-related factors, (3) infrastructure-related factors, (4) markets, (5) organizations and relationships, (6) environment, and (7) human resources and work environment.

Risks and costs have a very close relationship, especially in SS-LNG supply chains; for example, the more dangerous a shipping line is, the higher the costs of transportation and shipping insurance that must be borne [29,30]. In developing the SS-LNG supply chain, various risk factors must be considered to encourage an optimal LNG import portfolio, such as sea transportation distances, pirate attacks, import economic risks, export country political risks, and sea transportation risks [29]. Gurning [21] states that disruption due

to risk can cause high costs in the supply chain system as well as problems such as long waiting times, stock outs, the inability to meet customer demands, and increased costs.

The total cost in the LNG supply chain is divided into two main components, namely capital (Capex) and operational (Opex) expenditures. Capex is the entire initial investment cost incurred for constructing existing facilities in the collection terminal. The cost of investing facilities in these terminals also contain: (1) jetty facilities, (2) LNG offloading facilities, (3) cryogenic pipes, (4) storage tanks, (5) pumps, (6) vaporizers, (7) BOG compressors, (8) generators, and (9) building support and component installation [12]. Meanwhile, Opex is all costs incurred to support LNG distribution operations, including: (1) the operating costs of the collection terminal and (2) the transportation costs for shipping LNG from the refinery to the collection terminal. The operating costs also include: (1) ship rental costs, (2) ship fuel costs, (3) port costs, (4) collection terminal ship operating costs, (5) collection terminal electricity and fuel costs, (6) maintenance costs, and (7) collection terminal workers' payments [17].

Regarding the Capex, Pratiwi et al. [13] included the investment cost component for the construction of small-scale LNGc, jetty, LNG tanks, regasification units, and gas distribution systems. However, the operating and maintenance cost components of the collection terminal and LCT were included as Opex. Transportation costs are also used to calculate the expenses incurred regarding the operation of various transport modes, such as a ship, within a specific period [31–33]. Based on these conditions, the annual ship operating costs, such as the operational, shipping, capital, loading, and unloading expenses, are emphasized [31–34].

Based on Figure 4, the cost of each component and the supply chain used were observed and described. The total annual cost of each supply chain was also described against the selected LNGc capacity. In this case, the dotted lines denoted the partial solutions with the lowest cost. The scheme that used the LNGc greater than 41,000 m$^3$ produced an enormous total cost since renting LNGc made the cost of the LNG collection terminal infrastructure more significant. The cost was also high because the amount of LNG carried was similar to what was needed to meet the specified supply chain. Meanwhile, the scheme produced a smaller total cost when using LNGc < 41,000 m$^3$ because the cost required for LNGc rental increased two times for LNGc 21,000 m$^3$ as 2 units of 21,000 m$^3$ were required at the destination to replace the 41,000 m$^3$ capacity. These results also showed that the use of more LNGc also influenced the increase in the port berthing and canal costs. This shows the relationship between the supply chain model, the cost of each element, and the total expenditure required. Figure 4 shows the optimal supply chain models that used 1 LNGc (41,000 m$^3$) to serve all locations and generate the lowest annual price (Figure 4).

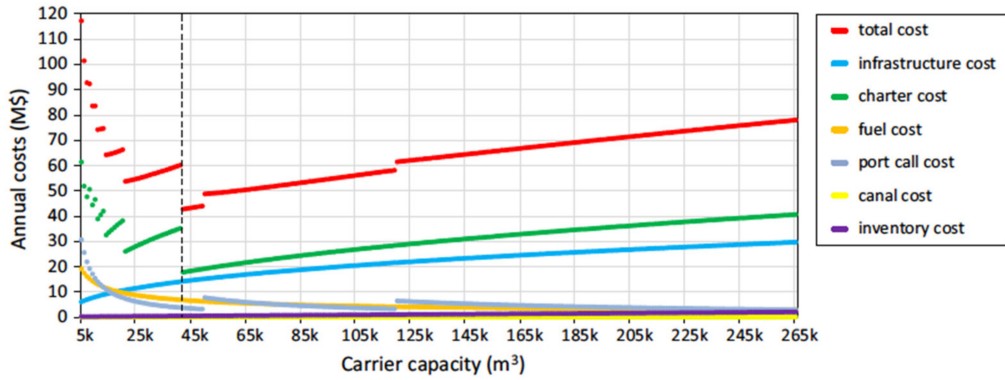

**Figure 4.** Cost Relationship and LNG Supply Chain Model [35].

## 3. Methods

The complexity of SS-LNG supply chain cases led to the collection of indicators through literature and in-depth interviews with experts [36]. Findings from the literature

and in-depth interviews were then analyzed using the Delphi approach to determine the risk indicators that best suit the characteristics of SS-LNG in Indonesia. The next step is to weight the possible risk factors in an SS-LNG supply chain using a risk index assessment, with a minimum of three levels of procedures [37]. Probability values were assessed with a range of 0–5, where 0 = 0% probability, 1 = 10% probability, 2 = 25% probability, 3 = 50% probability, 4 = 75% probability, and 5 = 100% probability [21]. Expert validation was used to answer questions that emphasized the following elements: (1) the patterns by which the supply chain conditions support the distribution of LNG to PLTG/PLTMG in the Indonesian archipelago, (2) the risks involved in the LNG supply chain activity, and (3) the relationship between risk indicators and SS-LNG supply chain variables. The population used in this study prioritized the practitioner and academic experts in the supply chain and science management sectors, respectively. Moreover, a non-random sampling method was used to select only 32 experts [38,39].

The use of the Delphi technique is intended to overcome the uncertainty space that often occurs due to the subjectivity of choice; it also incorporates statistical elements to guarantee the process test [40]. This technique has been developed in various disciplines, such as research on risk analysis models for shipping operational risks [40], bus safety [41], and transport infrastructure [42]. The criteria used for the provision of the validation experts are observed as follows:

- Practitioner Experts: (1) have at least 5 yrs of work experience and (2) occupy a managerial position in the supply chain sector for at least 5 yrs with a good reputation;
- Academic Experts: (1) have education and knowledge supporting a minimum academic level of bachelor S1 and (2) have at least 10 yrs of teaching experience with a good reputation.

Characteristics of the experts are shown in Table 2.

**Table 2.** Characteristics of informants (experts).

| No | Characteristics | Percentage (%) |
|----|-----------------|----------------|
| | Gender: | |
| 1 | - Male | 74% |
| | - Female | 26% |
| | Work experience: | |
| | - <5 years | 12% |
| 2 | - 5–10 years | 38% |
| | - 10–15 years | 34% |
| | - >15 years | 16% |
| | Education: | |
| 3 | - Bachelor | 63% |
| | - Master's/Doctoral | 37% |
| | Work unit: | |
| | - Operation and maintenance | 12% |
| 4 | - Engineering and planning | 47% |
| | - Business development | 19% |
| | - Academics | 22% |

There are two research stages, namely the Delphi analysis stage and the risk index assessment stage. Stage 1 begins with input from the literature and the results of expert brainstorming, which are then compiled in the form of a questionnaire to be validated in stages over two rounds. At the end of the validation, we selected influential risk indicators. The output from the Delphi stage became the input for risk assessment. The risk assessment is carried out by analyzing the possibility of risk, the impact of risk, and the risk index. The

results of this risk index assessment are used to mitigate the risks with the largest index. The complete research framework can be seen in Figure 5.

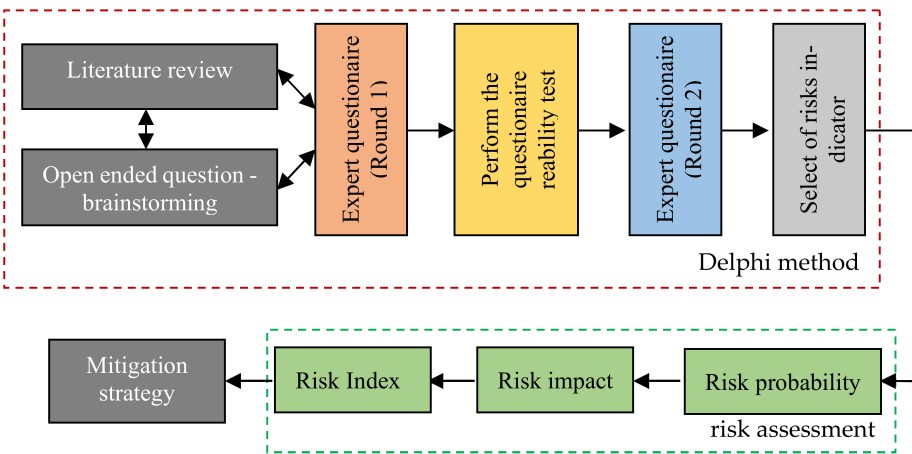

**Figure 5.** Research Framework.

## 4. Results

### 4.1. Validity and Reliability Test

The results of the validity test for all indicators show that the r calculated value compared to the r table value for *n* = 32 with a 95% (2-tailed) confidence interval is 0.349; therefore, the level of importance for each indicator is declared valid. The results of the reliability test show a 0.999 level of importance. If the calculated reliability value is greater than 0.6 (Cronbach Alpha Criteria), the indicator is considered reliable. Based on these results, the instrument used in this study has a very high or very good level of reliability (>0.9).

### 4.2. Determination of Risk Indicators Based on Expert Validation

The stages of determining risk indicators are validated against risk indicators based on all SS-LNG supply chain activities shown in Table 1. In round 1, 37 risk indicators that affect LNG supply chain activities on a small scale were found in the research findings from the literature (shown in Table 3 column c). A total of 18 experts had different perceptions of the five indicators, namely smuggling, spying, refugees, lack of railway facilities, and insufficient containers. They argued that the five indicators do not occur in SS-LNG supply chain activities in Indonesia. Round 2 was then performed based on these results to validate the five indicators. The results of the expert validation agreed that only 30 indicators could occur in the case of Indonesia (shown in in Table 2 column d). As a result, 30 risk indicators were chosen and further investigated. The findings of the variables used are summarized in Table 3.

**Table 3.** Variables related to Risk Indicators.

| Code | Sub-Variables/Indicators | Reference | In Depth Interview (Expert) |
|---|---|---|---|
| (a) | (b) | (c) | (d) |
| X.2.1 | Security and safety | | |
| X.2.1.1 | Ship accident | [21,28] | Yes |
| X.2.1.2 | Politics (riots and wars) | [21,28] | No |
| X.2.1.3 | Piracy | [21,28] | Yes |
| X.2.1.4 | Terrorist attack | [21,28] | No |
| X.2.1.5 | Sabotage | [28] | Yes |
| X.2.1.6 | Smuggling | [28] | No |
| X.2.1.7 | Spy | [28] | No |
| X.2.1.8 | Epidemic | [28] | Yes |
| X.2.1.9 | Refugee | [28] | No |

**Table 3.** *Cont.*

| Code | Sub-Variables/Indicators | Reference | In Depth Interview (Expert) |
|---|---|---|---|
| **(a)** | **(b)** | **(c)** | **(d)** |
| X.2.2 | Service-related factors | | |
| X.2.2.1 | Equipment damage | [21,25,28] | Yes |
| X.2.2.2 | Power outage | [21,25,28] | Yes |
| X.2.3 | Factors related to infrastructure | | |
| X.2.3.1 | Communication facility failure | [21,25,28] | Yes |
| X.2.3.2 | Lack of railway facilities | [21,25,28] | No |
| X.2.3.3 | Port Congestion | [21,25,28] | Yes |
| X.2.3.4 | Land access issues | [21,25,28] | Yes |
| X.2.3.5 | Limited storage capabilities | [25,28] | Yes |
| X.2.3.6 | Inadequate anchoring capability | [25,28] | Yes |
| X.2.4 | Market | | |
| X.2.4.1 | Bunkering costs uncertain | [21,28] | Yes |
| X.2.4.2 | Lack of transport ships | [21,25,28] | Yes |
| X.2.4.3 | Insufficient containers | [21,25,28] | No |
| X.2.4.4 | Inaccurate demand forecasts | [28] | Yes |
| X.2.5 | Organization and relationships | | |
| X.2.5.1 | Port strike | [21,28] | Yes |
| X.2.5.2 | Slow quarantine | [21,28] | Yes |
| X.2.5.3 | Old customs process | [21,28] | Yes |
| X.2.5.4 | Port shipping dispute | [21,28] | Yes |
| X.2.5.5 | Less flexible schedules drawn up | [25,28] | Yes |
| X.2.6 | Environmental | | |
| X.2.6.1 | Bad weather | [21,25,28] | Yes |
| X.2.6.2 | Earthquake | [21,25,28] | Yes |
| X.2.6.3 | Tsunami | [21,25,28] | Yes |
| X.2.7 | Human resources and work environment | | |
| X.2.7.1 | Lack of skilled workforce | [25,28] | Yes |
| X.2.7.2 | Lack of motivation | [25,28] | Yes |
| X.2.7.3 | Sailor's mental health | [25,28] | Yes |
| X.2.7.4 | Human error | [28] | Yes |
| X.2.7.5 | Unreasonable welfare | [28] | Yes |
| X.2.7.6 | Diversity of languages and cultures | [28] | Yes |
| X.2.7.7 | Poor safety culture | [25,28] | Yes |
| X.2.7.8 | Low level of safety leadership | [25,28] | Yes |

*4.3. Probability and Impact of Risk Indicators*

The frequency of the 30 risks that may occur in Indonesia's SS-LNG supply chain was determined to map the possible mitigation processes. Determination of event frequency is based on expert judgment and can be statistically calculated by looking at the mean value of the data. The specified frequencies are never (score = 0); once a year (score = 1); four times a year (score = 2); once a month (score = 3); twice a month (score = 4); and once a week (score = 5) [21]. The results of the analysis found that the most frequent probability of a risk occurring is once a year and four times a year, as shown in Table 4.

**Table 4.** Frequency of risk events in SS-LNG in Indonesia.

| Frequency | | Never | | Once a Year | | Four Times a Year |
|---|---|---|---|---|---|---|
| Risk Indicator | - <br> - | Piracy (0.41) <br> Sabotage (0.22) | - <br> - <br> - <br> - <br> - <br> - <br> - <br> - <br> - <br> - <br> - <br> - <br> - <br> - <br> - <br> - <br> - <br> - <br> - <br> - | Ship accident (0.72) <br> Epidemic (0.69) <br> Equipment damage (1.34) <br> Communication facility failure (1.19) <br> Inadequate anchoring capability (1.41) <br> Limited storage capabilities (1.38) <br> Bunkering costs uncertain (1.31) <br> Lack of transport ships (1.34) <br> Port strike (0.59) <br> Slow quarantine (1.09) <br> Old customs process (1.38) <br> Port shipping dispute (0.84) <br> Less flexible schedules drawn up (1.41) <br> Earthquake (0.97) <br> Tsunami (0.53) <br> Lack of skilled workforce (1.47) <br> Lack of motivation (1.34) <br> Sailors' mental health (1.41) <br> Diversity of languages and cultures (1.41) <br> Poor safety culture (1.47) | - <br> - <br> - <br> - <br> - <br> - <br> - <br> - | Power outage (1.56) <br> Port congestion (1.53) <br> Land access issues (1.56) <br> Low level of safety leadership (1.53) <br> Inaccurate demand forecasts (1.63) <br> Unreasonable welfare (1.56) <br> Human error (1.75) <br> Bad weather (2.13) |

Note: risk (mean value).

A total of 20 indicators have a probability of occurring once a year; 8 indicators may arise 4 times a year; as many as 2 indicators are never expected to occur. In the case of Indonesia, the experts are of the opinion that there is no risk of piracy and sabotage.

Once the frequency of occurrence is known, the probability of the risk indicator is calculated (in %), with 0–5 probability levels (where 0 = 0%; 1 = 10%; 2 = 25%; 3 = 50%; 4 = 75%; and 5 = 100%). The experts provide an assessment of how frequently a risk could occur, expressing it as a percentage. The probability of the financial impact that occurs on all risks based on costs (in IDR) is also calculated. Descriptive statistics for probability interpretation of maritime risk as applied by Gaonkar and Viswanadham (2007) and Handfield et al. (2007) are calculated. As with risk probabilities, the financial impact of risk is also divided into 0–5 impact levels (where 0 = no cost; 1 = IDR 1.5 billion; 2 = IDR 1.5–7.5 billion; 3 = IDR 7.5–15 billion; 4 = IDR 7.5–15 billion; 4 = IDR 15–75 billion, and 5 = IDR > 75 billion).

According to Gurning [21], the simultaneous calculation of risk probability and risk impact will be in the form of a supply chain risk index; in this study, the SS-LNG supply chain is in the form of a risk matrix. Research by Faisal et al. [43] and McCormack [44] in the wheat supply chain also shows a risk index for each stage of activity, so mitigation can be completed earlier in the process. Figure 6 shows that bad weather (29%) has the highest probability, followed by an inaccurate demand forecast (25%), human error (24%), bunkering cost uncertainty (23%), and equipment damage (22%). This finding proves that the influence of the weather on the SS-LNG supply chain is still very large, especially in the Indonesian archipelago, where the service locations are spread across small islands. This is in line with research by Humang et, al. [31], which found that the process of distributing goods by ship in Indonesia is constrained by weather and uncertainty in cargo.

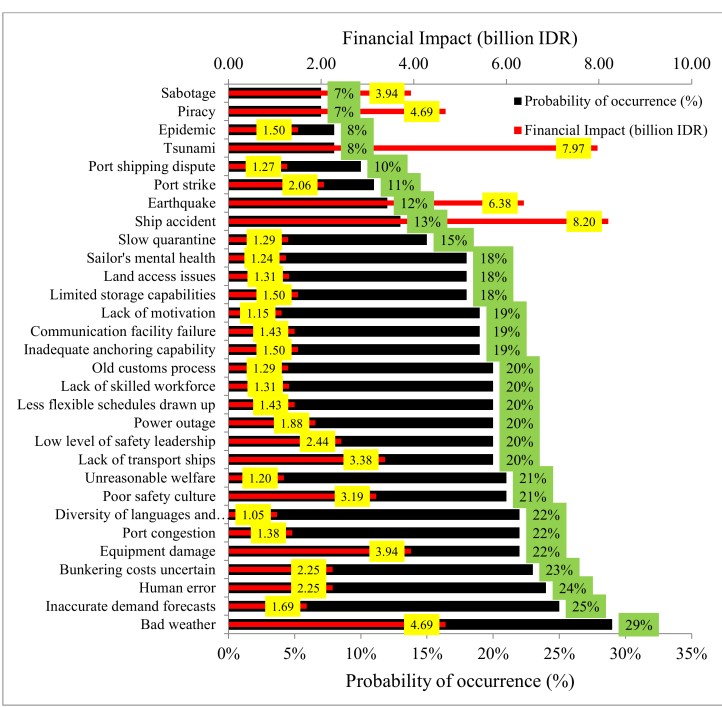

**Figure 6.** Probability and financial impact of SS-LNG supply chain risks in Indonesia.

When viewed in terms of the financial impact, the risk of ship accidents has the biggest impact, with a potential loss of IDR 8.20 billion, followed by tsunami risks (IDR 7.97 billion) and earthquakes (IDR 6.38 billion). Environmental and natural factors have the greatest impact, although the frequency of occurrence is very low. Other technical factors range from IDR 1.05 to 3.94 billion. Even though the technical factor has a lower financial impact, the frequency of occurrence is likely to be greater, so it will have a significant effect on the SS-LNG supply chain.

### 4.4. SS-LNG Supply Chain Risk Index

Based on the risk probability and financial impact that may occur, the risk index for all risk indicators can be explained. The greater the risk index, the higher the probability of occurrence and the resulting financial impact. The risk index is the product of the multiplication of the risk probability index and the financial impact index [43,44]. Based on the 30 risk indicators in the risk index assessment, the 9 largest index risks were determined, as shown in Table 5.

**Table 5.** The highest order of the SS-LNG supply chain risk index.

| Code | Risks Indicator | Frequency | Probability | | Impact | | Risk Index |
|------|-----------------|-----------|-------|---------|-------|---------|------------|
| | | | Index | Average | Index | Average | |
| (a) | (b) | (c) | (d) | (e) | (f) | (g) | (h) = (d × f) |
| X.2.6.1 | Bad weather | 4 times a year | 3.3 | 29% | 3.0 | 4.69 | 9.9 |
| X.2.1.1 | Ship accident | 1 times a year | 2.3 | 13% | 3.6 | 8.20 | 8.2 |
| X.2.2.1 | Equipment damage | 1 times a year | 2.8 | 22% | 2.9 | 3.94 | 8.2 |
| X.2.7.4 | Human error | 4 times a year | 2.9 | 24% | 2.6 | 2.25 | 7.6 |
| X.2.4.3 | Inaccurate demand forecasts | 4 times a year | 3.0 | 25% | 2.5 | 1.69 | 7.6 |
| X.2.7.7 | Poor safety culture | 1 times a year | 2.7 | 21% | 2.8 | 3.19 | 7.6 |
| X.2.4.2 | Lack of transport ships | 1 times a year | 2.7 | 20% | 2.8 | 3.38 | 7.6 |
| X.2.4.1 | Bunkering costs uncertain | 1 times a year | 2.9 | 23% | 2.6 | 2.25 | 7.5 |
| X.2.7.8 | Low level of safety leadership | 4 times a year | 2.7 | 20% | 2.7 | 2.44 | 7.1 |

The biggest risk index according to Table 5 is bad weather with a value of 9.9, followed by a ship accident and equipment damage of 8.2, and human error, inaccurate demand

forecasts, poor safety culture, and a lack of transport ships all having a value of 6.7. After obtaining the risk index, all risks are mapped and shown in Figure 7.

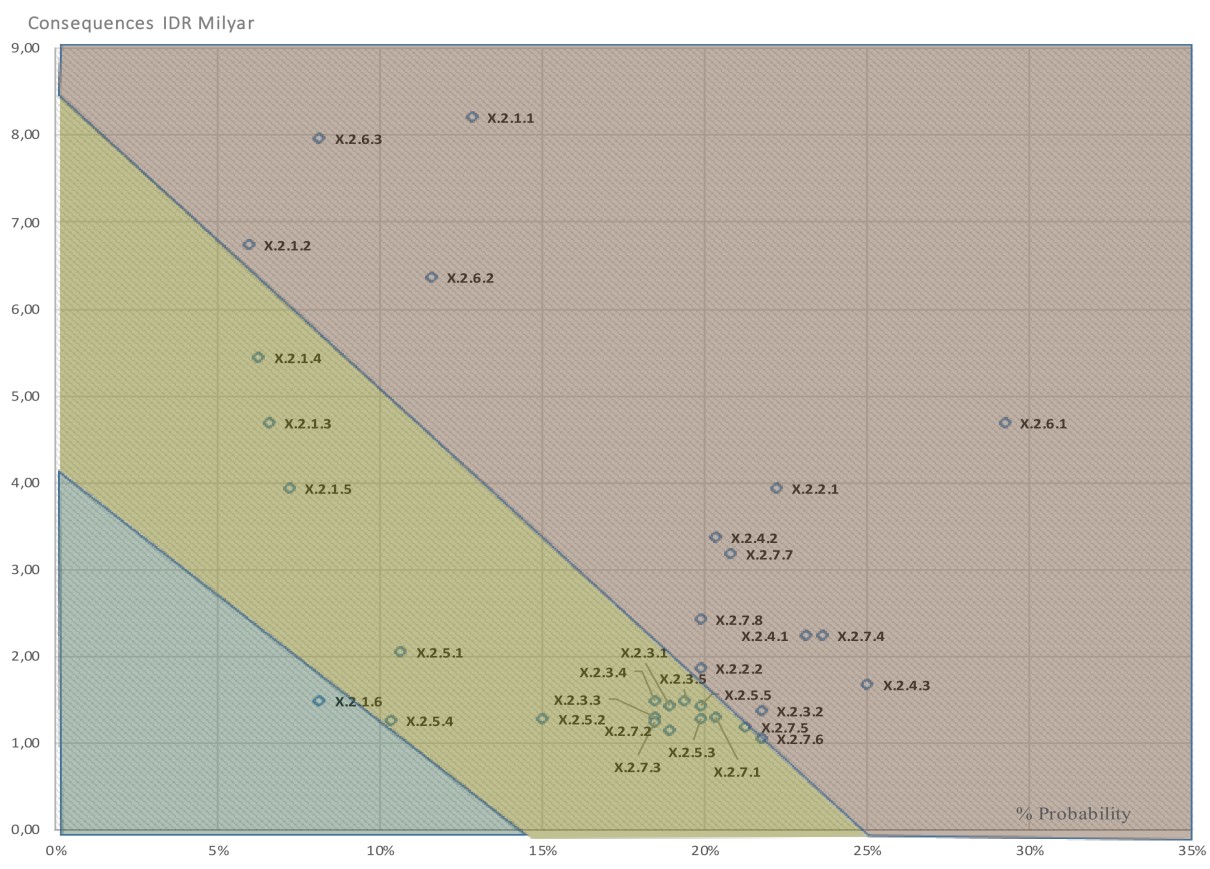

**Figure 7.** Classification and Rating of Risk Indicators by Disturbance Level.

Note :

| **Low disruption risk events** | | **Medium disruption risk events** | | **High risk events** | |
| --- | --- | --- | --- | --- | --- |
| X.2.1.6 : Epidemic | 1 | X.2.3.5 : Limited storage capabilities | 1 | X.2.6.1 : Bad weather | 1 |
| X.2.5.4 : Port Shipping Dispute | 2 | X.2.3.4 : Land access issues | 2 | X.2.1.1 : Ship Accident | 2 |
| | | X.2.5.5 : Less flexible schedules drawn up | 3 | X.2.2.1 : Equipment damage | 3 |
| | | X.2.3.1 : Communication facility failure | 4 | X.2.7.4 : Human error | 4 |
| | | X.2.7.1 : Lack of skilled workforce | 5 | X.2.4.3 : Inaccurate demand forecasts | 5 |
| | | X.2.5.3 : Old Customs Process | 6 | X.2.7.7 : Poor safety culture | 6 |
| | | X.2.3.3 : Land access issues | 7 | X.2.4.2 : Lack of Transport Ships | 7 |
| | | X.2.7.5 : Unreasonable welfare | 8 | X.2.4.1 : Bunkering costs uncertain | 8 |
| | | X.2.5.2 : Slow Quarantine | 9 | X.2.6.2 : Earthquake | 9 |
| | | X.2.7.3 : Sailor's mental health | 10 | X.2.7.8 : Low level of safety leadership | 10 |
| | | X.2.5.1 : Port Strike | 11 | X.2.2.2 : Power outage | 11 |
| | | X.2.7.6 : Diversity of languages and cultures | 12 | X.2.3.2 : Port Congestion | 12 |
| | | X.2.7.2 : Lack of motivation | 13 | X.2.6.3 : Tsunami | 13 |
| | | X.2.1.4 : Terrorist Attack | 14 | X.2.1.2 : Politics (riots & wars) | 14 |
| | | X.2.1.3 : Piracy | 15 | | |
| | | X.2.1.5 : Sabotage | 16 | | |

From the risk index assessment, the risk categories are divided into low risk, medium risk, and high risk, as shown in Figure 7. The low-risk category contains 2 risk indicators, the medium-risk category contains 16, and the high-risk category contains 14. Each of these risk indicators will be examined further to determine the extent to which the relationship between them affects SS-LNG supply chain activities.

### 4.5. Discussion and Mitigation Strategies

Based on the risk index, a mitigation strategy is developed that consists of four mitigation criteria: inventory and resource mitigation; contingency route changes; business continuity planning; and recovery planning, which can be applied in the LNG supply chain from the elements of the LNG plant, transportation, and LNG receiving terminal. Gurning's research [21] and an expert validation survey found that only 16 risk mitigation strategies can be applied to SS-LNG supply chain activities, from LNG loading activities to gas distribution to end users. Three activity groups can be determined from the LNG supply chain activities in Table 1, comprising: the first LNG loading activities both at the LNG refinery and LNG receiving terminal; both LNG shipping and picking activities; and LNG storage activities (LNG regasification and gas distribution). Five major risk mitigation strategies were found in each activity group based on the number of responses from experts, and with the proportion of recommendations for the five mitigation strategies approaching the same proportion [45,46].

In the first group, LNG loading activities related to shipping risk indicators that have the most impact are bad weather, ship accidents, damage to ship equipment, earthquakes, and tsunamis, which can delay, increase the duration of, and even stop the LNG loading process. In this study, it was found that the best risk mitigation strategy recommendation is business continuity planning by changing work practices so that risk mitigation activities that must take place prior to the risk occurring should agree on a force majeure clause and the rights and obligations of all parties. The transportation cost clause is included in the LNG price formula and the Annual Delivery Program (ADP) clause in the agreement (TCP, SLA, SPA). Risk mitigation activities can include the philosophy of spares or redundancy of existing equipment on board, including replacement vessels and the Shop Shore Compatibility Study (SSCS). When a risk event occurs, options include reducing daily LNG production or altering the refinery operation pattern.

In the second group, LNG shipping and retrieval activities that have the most impact on risk indicators are bad weather, ship accidents, shortages of transport vessels, poor safety cultures, and tsunamis, which can delay transportation, increase the duration of the voyage, and even stop the process of sending and taking LNG. This study found that the best risk mitigation strategy recommendations include business continuity planning, where changes to work practices are made so that risk mitigation activities that must be completed prior to a situation occurring agree on a force majeure clause and the rights and obligations of the relevant parties for each activity. The transportation cost clause is included in the LNG price formula and the ADP clause in the agreement (TCP, SLA, SPA). Potential risk mitigation activities include using the philosophy of spares and redundancy of equipment on board, replacement ships, developing skill and training programs for workers, and developing standards and operating procedures. When a risk event occurs, daily LNG production may be reduced or the pattern of refinery operations may be changed.

In the third group, LNG storage and gas distribution activities, the most impactful risk indicators are equipment damage, earthquakes, poor safety culture, tsunamis, and low levels of safety leadership, which can delay production, damage facilities, and even stop the LNG storage and distribution process. gas. According to the findings of this study, the best risk mitigation strategy recommendation is business continuity planning, where work practices are changed so that risk mitigation activities that must be completed before a risk event occurs agree on a force majeure clause, the rights and obligations of the parties for each activity, and ADP Clauses in agreements (TCP, SLA, SPA). Risk mitigation activities include using the philosophy of equipment spares and redundancy at the LNG receiving terminal, formal pre-assignment assessment, competency standardization, a development (skill) and training program for workers, and developing standards and operating procedures. When a risk event occurs, it may require a reduction in daily LNG production or a change in the pattern of refinery operations. Details of risk mitigation activities for each risk indicator, which are the top five risk indicators in each LNG supply chain activity, are presented in Table 6.

**Table 6.** Mitigation Activities in SS-LNG Activities Based on Risk Indicators.

| Risk Mitigation | | Risk Indicator |
|---|---|---|
| **Strategy** | **Activity** | |
| Changes in work practices | 1. Agree on the risk mitigation clause in the agreement (a. force majeure clause, rights and obligations of the parties for each activity, including preparing a replacement ship in the agreement; b. transportation costs are included in the LNG price formula; c. contract term clause; d. clause for the annual delivery program (ADP)).<br>2. Employ a security team and a patrol team (SEA and ROW).<br>3. Reducing daily LNG production (by changing the operating patterns of refineries and LNG receiving terminals)<br>4. Implement human resource strategy (a. formal pre-assignment assessment; b. competency standardization; c. standardizing grade, salary, compensation, and benefits according to the job description; d. scheduling rotation on/off shipmen; e. conducting routine health assessments; f. implementing a development program (skills) or training for workers).<br>5. Develop standards and operation and maintenance procedures.<br>6. Ship-Shore Compatibility Study | - Ship Accident<br>- Equipment breakdown<br>- Lack of Transport ship<br>- Bad weather, Earthquake and Tsunami<br>- Poor safety culture and Low level of safety leadership |
| Develop an early warning system | 1. Placement of "Buoys" around the wharf sea area<br>2. Installation of the radar system in the "basin<br>3. Development of weather forecasts<br>4. Develop an alarm system. | - Ship Accident<br>- Bad weather, Earthquake and Tsunami |
| Alternative route | 1. Develop alternative shipping lanes or berthing ports.<br>2. Monitoring the movement of LNGc and support ship | |
| Alternative planning | Develop and implement:<br>1. Security and safety strategy,<br>2. Equipment spare/redundancy philosophy in the design and operation phases | - Equipment damage |
| Order Policy Optimization | Agree on optimal planning and scheduling clauses in the sales and purchase agreement (SPA) | - Lack of Transport Ship |
| Supply Flexibility | Agree on ADP Clauses, Spot Cargo Options, Multi-Source LNG, SWAP Mechanism, and Substitute Fuel in Agreement. | |

In the final validation stage, the expert gave an "agree" opinion on the findings above, responding that the dominant risk is the maritime risk associated with shipping activities, including from the loading, unloading, shipping, and retrieving LNG that occurs at the LNG plant up to the terminal. One expert also explained that one of the reasons that the human factor is dominant in supply chain activities at the LNG receiving terminal is LNG storage and gas distribution activities, as they have lower safety standards than gas regasification and distribution activities. The following is the response of the LNG supply chain scheme expert in the final expert validation:

> Agree, special risks frequently arise on ships, in old berths, or when ships are not docked. So that often happens, and our ship explodes with steam. If there is a shortage of transport ships, it is actually because the ships are damaged, so it is as if there is a shortage of ships. But that can be mitigated in the scheduling of cargo when dry dock occurs; when it has to be replaced with another ship only if the case is due to an accident so that a replacement ship is needed, it is difficult to mitigate. What can still be mitigated is the shortage of transport vessels, in the sense that there is a significant increase in demand or there is a supply chain failure at PLN, for example:

> "Head of Commercial Division of Mid-stream LNG Company."

Table 6 shows the mitigation strategies and risk mitigation activities have been prepared for the top five risks that have received expert final validation.

In the expert's final validation stage, they also explained several details of risk mitigation activities that occurred when LNGc monitoring went through the satellite and provided notification to the terminal; it did not arrive at the receiving terminal for between 20 h and 9 h. The ADP activity is designed to synchronize the production plan from the LNG plant to the end user for the next year; this is agreed upon at the end of the previous year. Transportation will be arranged so that all times are understood, such as when the LNGc ship will arrive,

when the dry dock will be known, how much time it will be there, and when a replacement ship should be prepared. If an ADP ship accident occurs, this can serve as a guideline for how quickly a replacement ship should arrive, whether the ship should be repaired or a replacement ship should be found, and how to respond to mitigation related to changes in operating patterns at refineries and LNG receiving terminals.

## 5. Conclusions

The SS-LNG supply chain in the Indonesian archipelago starts from the loading process at the LNG plant and ends with the distribution of gas to the end user. Expert validation found that the SS-LNG supply chain includes 8 activities that have 30 potential risk indicators. Out of the 30 risk indicators, 9—namely bad weather, ship accidents, equipment damage, human error, inaccurate demand forecast, poor safety culture, lack of vessels to carry out the transport, earthquakes, tsunamis, and low levels of safety leadership—are considered to have a higher risk index.

Regarding probability of occurrence, the top three events are bad weather (29%), inaccurate demand forecasts (25%), and human error (24%). In terms of financial impact caused, ship accidents rank first, with a potential loss of IDR 8.20 billion, followed by natural events such as tsunamis (IDR 7.97 billion) and earthquakes (IDR 6.38 billion); even though they occur very infrequently, their impact is high.

To optimize the SS-LNG supply chain process, several approaches can be taken, including changing work practices, developing early warning systems, optimizing routes and backup or alternative plans, optimizing separation policies, increasing supply flexibility, and developing weather warning systems. The limitation of this research is that the expert's judgment is very subjective in assessing the risk index, so validation must be carried out repeatedly. From the results of this paper, further research can be carried out by analyzing how much influence each risk has in terms of additional costs (operational, investment), shipping time, and length of routes navigated in the process of supply chain operations. The results can help produce an optimal risk-based supply chain model in areas with archipelagic characteristics in Indonesia.

**Author Contributions:** Conceptualization, R.A.M., N.W. and W.P.H., methodology, R.A.M., W.P.H. and Y.Y.A.P.; validation, R.A.M., W.P.H., N.W. and Y.Y.A.P.; formal analysis, W.P.H., N.W. and I.K.; investigation, R.A.M., I.K. and Y.Y.A.P.; resources, R.A.M., N.W., I.K., W.P.H. and Y.Y.A.P.; data curation, I.K. and Y.Y.A.P.; writing—original draft preparation, R.A.M., N.W. and I.K.; writing—review and editing, N.W., W.P.H. and Y.Y.A.P.; visualization, N.W. and Y.Y.A.P.; supervision, I.K and W.P.H. All authors have read and agreed to the published version of the manuscript.

**Funding:** The author would like to acknowledge the support of this work provided by the Directorate of Research and Development, Universitas Indonesia, under the Riset Kolaborasi Indonesia Program (RKI) 2022 (Contract No NKB-1064/UN2.RST/HKP.05.00/2022).

**Data Availability Statement:** The data presented are available upon request from the corresponding author.

**Conflicts of Interest:** The authors declare no conflict of interest.

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
