# Peer review of "Probability of Risk Factors Affecting Small-Scale LNG Supply Chain Activities in the Indonesian Archipelago"

_infrastructures, doi:10.3390/infrastructures8040074_

Round 1

Reviewer 1 Report

1. Literature review is weak for this paper

2. In the introduction section, the authors should better highlight the objectives of their work and to what extent it contributes to closing the gap in the existing literature and/or practice. The innovative value of the contribution should be particularly highlighted.

3. In the introduction section, the authors should provide more thorough information about the existing model and emphasize its advantages and weaknesses.

4. This application topic has not received much attention in the literature. However, the study, literature review, and presentation require a substantial improvement in several respects.

5. Result and Discussion sections are inadequate. Need more attention and a better explanation.  The conclusion section is a bit too sparse. The authors should emphasize the impact and insights of the research and provide several solid future research directions. Also, the limitations of the model should be mentioned.

Author Response

Thank you for your input.
We have corrected this paper, starting from correcting the title, objectives, and analysis.

best regards

Reviewer 2 Report

This manuscript analyzed the risk factors affecting small-scale LNG supply chain activities in Indonesia using the Delphi analysis method, but there are insufficient parts in the format of the manuscript and the contribution of the study, such as problems with the number system.

The specific parts in need of improvement are as follows.

1. It is recommended to revise the manuscript title including the method of the study.

2. As the readability of all figures are poor, please modify them to a clearer and more readable figure.

3. The description of each figure and table should be in the manuscrpit, but it is insufficient.

- For example, the description of figure 1 is on line 41 of page1, but there is no description of figure 4.

4. There is a problem with the figure numbering system of the manuscpit.

- For example, there should be figure 5 after figure 4, but there is no figure 5, but figure 6.

5. The purpose of this study is stated on line 171 of page 5, but it is necessary to mention and indicate the research method so that a more specific purpose and differentiation from existing studies can be clearly revealed.

- Meta-analysis has not been conducted in previous studies, but it is necessary to descript in more detail why this method is more suitable.

6. It is necessary to unify the terms used in the manuscript.

For example, Ship on page 5 196, vessels on page 6 197

7. Please check the units on line from 220 to 222 numbers and correct them appropriatly.

For example 30,000m3 not 30.000m3

8. Please check terminology on the manuscript and modify it appropriatly.

For example, rental costs in line 226 needs to be changed to a technical term such as charterage.

9. There is a problem with the numbering system of the manuscrpit.

For example, 2.2 should come after 2.3, but 2.5 comes out and

chapter 4 should come after chapter 3, but chapter 5 comes.

10. Manuscripts should be fully edited for readability.

For example, table 3 mentioned on page 8 at line 225 is mentioned at line 383 on page 11, making it difficult to read.

11. The part about the qualifications of experts is mentioned in line from 365 to 379, but

- How many people did you survey?

- What is the basis for such a number of people?

- There is no mention of whether validity and reliability tests were conducted in the questionnaire for conclusions.

12. Although the results and conclusions of this study are presented,

- There is no clear description of the contributions and limitations of this study according to the research results.

Author Response

(The authors gave the same response as above.)

Reviewer 3 Report

the paper proposed deals with the topic of Small-Scale LNG Supply Chain in the Indonesian Archipelago. The article is timely, due to shortage of supply in some countries.

my main concerns, apart a few shortcomings on the pagination that i will discuss later on, is that the article as it is now is more in a case-study / report form with a clear focus on a "local" perspective and does not provide insight and advancement to the scientific discussion. in addition the bibliography is rather poor, both in describing the LNG logistic chain and in discussing and proofing the methodology adopted. The methodology is merely a review and discussion of expert opinion and it is not detailed adequately.

The tables are not clear, not explained in the text and difficult to read. for example, in appendix A , what is the difference between items (i.e. X.2.1.4) accepted with all yes and cases (i.e. X.2.4.1 and others) that are accepted as well but with only 1 one positive remarks?  

as for the pagination, there are two section "2" . In the section "2 - theoretica study" the subparagraphs do not follow a sound numerical order. finally, table 3 is mentioned in the text on page 8, but the table is on page 11

Author Response

(The authors gave the same response as above.)

Round 2

Reviewer 1 Report

Accept as it is

Author Response

Comments on reviewer's input attached (pdf)

Reviewer 2 Report

The proposed research is interesting in general. However, the manuscript requires some major revisions, which to enhance its value to the research and professional community. They are presented in details below:

 1. Structure of the manuscript

In general, the structure of the study should clearly present the background of the study, analysis of previous research, purpose, method, contribution, and limitations. However, in this manuscript, the background of the study and previous research are prominently described, but the purpose, contribution, and limitations of the study are not clearly indicated.

Moreover, the numbering system on the manuscript is also inconsistent, resulting in significant flaws in the readability and structure of the manuscript.

For example, 2.2 should come after 2.1. However, in this manuscript, 2.5 is described after 2.1.

2. Purpose of the study

This manuscript is a study on risk factors for small scale LNG supply chain in Indonesia Archipelago. However, I can't clearly identify the purpose of this study anywhere in Introduction chaper. In addition, the Introduction must clearly describe how the approach was taken to achieve this purpose.

3. Research method of the study

This study was based on the results of interviews with 32 experts. The composition of experts is described in Table 2, but the research framework is shown incompletely in Figure 6 and is not described in detail in the manuscript. Therefore, it is not possible to confirm the method of specific study in this manuscript.

4. Limitations of the study

The limitations of this study should be recognized and follow-up studies to improve them should be clearly described in the conclusions. Follow-up studies are shown on the Conclusion chapter, but limitations of the study cannot be identified.

5. Reference

This study is based on a number of references. However, references in [1] are described in six manuscripts. Also, the references are not listed sequentially in the paper and are scattered in several places. This greatly reduces the readability of the paper.

Author Response

(The authors gave the same response as above.)

Reviewer 3 Report

i recognize the efforts by the authors to improve the paper according to my comments and the comments by other reviewers

improve the quality of fig 1 and 2

on page 9, a separation line is missing between the end of paragraph 1 and the beginning of paragraph 2. on the same page, the first linea are in bold, which is not needed. lines 316 and following need to be aligned

table 2 improve the quality of the table .. the margins are not aligned

figure 6 is difficult to read .. what about the text in the orange and grey boxes at the bottom? it is Delphi method (not "metode")

on page 10, lines 367 and 373 refer to table 2 .. i think that authors mean Table 3 instead. items X.2.1.1 and following should have a indentation with respect to item X.2.1 .. the same goes for the rest of the table

reformulate lines 366-368. "In round 1, the literature review highlighted 37 risk indicators (see table 3, column c), which affect LNG supply chain activities on a small scale".

do the values assigned to the assessment scales used in par 2 and 3.3 have any reference in the literature? 

in table 4, none of the risk factors falls into categories 3-4-5 if I have understood it correctly? can the authors explain why, in their opinion?

reformulate lines 392-395 pg 12 "20 indicators show a probability of  occurring once a year, 8 show a probability of occurring four times a year, and 2 risk indicator have are not likely to occur as a whole. In the case of Indonesia, experts agree in saying that piracy and sabotage never occurred".

page 10 line 401-402 .. why are the 2 references not in the short form between brackets [...]?
line 418 "as for the financial impact caused, it turns out that the risk of ship accident is the highest impact, with a potential loss of IDR 8.20 billion ... " is it better?

page 13 line 459 insert a line after the table

the labels on the axles in fig 8 have different size/font. i would improve english there 

improve the conclusion at operation and policy levels .. now the paragraph is too short and basically a summary of the section above

check the numbering of the reference list

Author Response

(The authors gave the same response as above.)

Round 3

Reviewer 2 Report

Dear Author,

The proposed research is interesting in general. At the same time the manuscript requires some revisions and adjustments, which to enhance its value to the research and professional community. They are presented in details below:

Abstract

Most of the content is about the background explanation. Please edit to include background, purpose, methods, results, and research contributions (research gab, including limitations).

Introduction

Most of the background information is in the introduction. Please state more clearly the purpose of the study and the research method mentioned in the second half of the introduction.

Lines 37-40

Insert references for sentences.

Line 51

Please indicate the full name of HSD at the first mention.

Line 64

Both US.nD/BOE and USD/BOE are mentioned. It appears to be mistyped, so please correct it.

Line 81

Mark 4-5m exactly as 4-5meters. Write subsequent units in full form without omissions.

Lines 113, 126, 192, 246

The terms SSLNG, SS-LNG, small-scale LNG, and SC-LNG supply are used interchangeably. Please use unified terminology

Methods

1. Clearly describe the reason for using analysis method, the delphi technique.

2. Investigate previous studies and describe it that this analysis technique is suitable for the manuscript.

Results

1. Present reliability and validity results such as table.

2. Investigate previous studies and describe that reliability and validity are appropriate.

Line 335

Check whether references are referenced, and if so, mark them in the appropriate reference mark '[ ]' and reference.

General revision of the manuscript

Spaces (Line 128), misspellings (Line 233, Line 236), terminology (Line 86, sea accident or marine accident), capitalization, font, reference references (Line 102, Line 257), manuscript numbering system (Chapter 2 It seems that a unified application for repeated) is needed.

Author Response

Thank you for your input,
we have improved according to your suggestions, and there are a number of things that we think are appropriate, for example we did not include research limitations in the abstract but in the conclusions.

But in general we have tried to improve according to your suggestions.

regards

Reviewer 3 Report

with respect to the earliest version, the shortcomings pointed out have largely been addressed.

while going through the paper, the structure should be refined a bit to improve the final product. my suggestion is:

1 - introduction = too long, the aim and scope of the paper is restricted to the last ten lines. i suggest a shorter paragraph of no more than 50 lines where aim and scopes, advances against previous literature and structure of the paper are highlighted

2 - context and literature background = here the authors can summarize what has been moved from par. 1 + the lng supply chain + the cost + the methodology ... WATCH OUT, THERE ARE STILL TWO SECTION #2 IN THIS VERSION OF THE PAPER .. i suggest merging them

3 - results = section 3.1 3.2 3.3 and 3.4

4 - discussion and mitigation strategies = section 3.5  

5 - conclusion = I am still a bit confused about this one, expecially by comparing the text and table 5 ... i suggest a new version below

===

The SS-LNG supply chain in the Indonesian archipelago starts from the loading process at the LNG plant AND ENDS WITH the distribution of gas to the end user.

EXPERT VALIDATION found OUT that THE SMALL SCALE LNG SUPPLY CHAIN IS COMPOSED OF 8 ACTIVITIES, FOR A TOTAL OF 30 risk indicators that MIGHT POTENTIALLY HAVE AN INFLUENCE. OUT OF THE 30 RISK INDICATORS, 9 - NAMELY bad weather, ship accidents, equipment damage, HUMAN ERROR, INACCURATE DEMAND FORECAST,  poor safety culture, lack of vessels TO CARRY OUT THE TRANSPORT, earthquakes, tsunamis, and low levels of safety leadership - ARE INDICATED AS THE ONES HAVING HIGHER RISK INDEX.

The TOP3 EVENTS AS FAR AS PROBABILITY OF OCCURRENCE IS CONCERNED ARE bad weather (29%), inaccurate demand forecasts (25%), and human error (24%). IN TERMS OF financial impact caused, ship accidents RANKS FIRST with a potential loss of IDR 8.20 billion, followed by NATURAL EVENTS SUCH AS tsunami (IDR 7.97 billion) and earthquake (IDR 6.38 billion), WHICH have great impact, although theIR frequency of occurrence is very low.

[etc etc ... ]

===

as for the last sentence (shortcomings and scope for further research) , I agree that expert validation is highly dependent on the subject evaluation of the experts .. personally, i would have given higher frequency of occurrence to VOLATILITY OF BUNKERING COSTS (are prices adjusted on a monthly basis or am i mistaken?) and LACK OF VESSELS FOR CARRYING OUT THE TRANSPORT (is also punctuality and reliability enclosed in this risk factor?!?) please provide clarification 

Author Response

thank you for the correction, we have fixed it according to your suggestions.
